# Alzheimer’s Disease Caregiver Characteristics and Their Relationship with Anticipatory Grief

**DOI:** 10.3390/ijerph18168838

**Published:** 2021-08-22

**Authors:** Alba Pérez-González, Josep Vilajoana-Celaya, Joan Guàrdia-Olmos

**Affiliations:** 1Faculty of Psychology and Education Sciences, Universitat Oberta de Catalunya (UOC), 08018 Barcelona, Spain; 2Department of Clinical Psychology and Psychobiology, Faculty of Psychology, University of Barcelona, 08035 Barcelona, Spain; 3Alzheimer Catalunya Fundació, 08006 Barcelona, Spain; jvilajoana@copc.cat; 4Department of Social Psychology and Quantitative Psychology, Faculty of Psychology, University of Barcelona, 08035 Barcelona, Spain; jguardia@ub.edu; 5Institute of Neuroscience, University of Barcelona, 08035 Barcelona, Spain; 6Institute of Complex Systems, University of Barcelona, 08028 Barcelona, Spain

**Keywords:** anticipatory grief, caregiver, Alzheimer’s, dementia, family members

## Abstract

In Alzheimer’s disease, two fundamental aspects become important for caregivers: ambiguity and ambivalence. Thus, anticipatory grief is considered an active psychological process that is very different from the mere anticipation of death. The present study aims to determine which characteristics of family caregivers of people with dementia, such as age, gender, educational level, relationship with the person with dementia, years with dementia or years as a caregiver, are related to the presence of anticipatory grief. A cross-sectional design was employed. The sample consisted of a total of 129 subjects who cared for a family member with dementia. A sociodemographic data sheet and a battery of tests measure the presence of anticipatory grief, caregiver burden and/or psychopathology. The results obtained allowed us to confirm some of the hypotheses regarding the anticipatory grief construct, the importance of the care time factor, in years and per day, as well as the relevance of the previous demographic and psychopathological profile (being female, spouse function and possible depressive symptomatology). Likewise, from the prediction analyzes performed, it seems that these variables can predict anticipatory grief. These results propose interesting opportunities to formulate care proposals to professionals and family caregivers in relation to care tasks and caregiver skills.

## 1. Introduction

Grief is related to a great variety of losses, which can be real or symbolic, physical, relational, functional, etc. In the same way, losses are not always clear and well determined. In the case of anticipatory grief, it is an emotional response to an expected and inevitable loss, which begins before it occurs, and allows the necessary readjustments until such time as it occurs [1].

Although there is still little research on ambiguous loss, it is always considered stressful and often torments [2]. There are two basic types of ambiguous loss [3]. In the first, the family members perceive a certain person as physically absent but psychologically present since it is not certain whether the person is alive or dead. This would be the case in disappearance or for contenders in armed conflicts. In the second type of ambiguous loss, the person is perceived as physically present but psychologically absent. People with Alzheimer’s disease illustrate extreme cases of this condition [3]. Uncertainty prevents people from adapting to the ambiguity of their loss by reorganizing the roles and norms of their relationship with loved ones. In this way, relationships between a couple or family freeze in that situation or evolve into conflicting or pathological patterns.

In addition to the question of ambiguity, a central aspect in the relationships between family caregivers and their relatives with dementia is ambivalence [1,4]. In the case of dementia, because it is a ‘nondefinitive’ loss that involves a very high and continuous dedication, the relationships are characterized by an intense ambivalence between the pain related to the person with dementia decline and the anger over the losses in the caregiver’s life, which are affected by the care task. In this sense, the characteristics of the dementia process itself, with multiple losses and at different levels, favor the experience of anticipatory grief in caregivers, as a specific feeling of grief before the death of the family member [5].

Ambiguity and ambivalence experienced by caregivers of people with dementia are two powerful stressors that can seriously interfere with the care task [6]. Several investigations highlight that the more a caregiver perceives their relative as psychologically absent, the less competent they feel, the greater their feeling of burden and the more emotional distress they experience [6,7]. Similarly, the bibliography shows that unrecognized emotional ambivalence can fuel negative relationships and destroy the resilience of the caregiver [8]. More specifically, some studies conclude that the presence of ambivalent feelings contributes significantly to the explanation of depressive and anxious symptoms in caregivers after controlling for sociodemographic and stressful variables [9]. 

In relation to the above, several researchers have studied the variables that influence the course of grief and simultaneously established predictors of complicated grief [10,11,12]. Although grief is considered a natural adaptive process in the face of a significant loss, its success is not always guaranteed. The variables, in general, can be grouped into individual (subject–lost object), family, relational, situational or circumstantial [13]. At the same time, it is possible to differentiate between risk and protective factors. Among the former, it is worth highlighting affective dependence, expressions of anger and guilt, prior psychic vulnerability (past psychopathology or unresolved past bereavements), feeling of loss of control and economic problems [10]. In contrast, protective factors are the ability to make sense of an experience, competencies in the management of diverse situations and the management of emotions and a good capacity for self-care [10]. 

Other studies have focused on the determinants or major drivers in the grieving process [14,15,16]. In terms of the context, the type of death, the cause and the long or short time between the ‘announcement’ to the event will be decisive. Generally, sudden or traumatic death complicates grief.

The type and quality of the relationship, as well as the kinship with the person with dementia, are very relevant variables for the course of grief. Relationships based on certain independence provide protective elements. Contrary to what is foreseeable, ambivalent or conflictive relationships do not favor a good mourning process but rather involve feelings of guilt and anger against the deceased [17]. 

The strongest and longest emotional reactions of grief are manifested when death is unexpected [12]. However, the progressive decline of relatives does not guarantee uncomplicated grief. It is recognized that the impact of a very serious diagnosis produces a change in the structure and functionality of the family system [18]. There are studies in oncological processes and mourning where the losses refer to the sick person or to his relatives [19], giving rise to the concept of anticipatory grief. However, few studies refer to processes of caregivers of people with dementia, where the decline of people is episodic or progressive [20,21,22].

Dementia processes such as Alzheimer’s disease are processes of a long evolution, generally between 8 and 10 years, with exceptions up to 20 years or more, which subjects caregivers and family members with dementia to a high and progressively dependent relationship. In this sense, the task of caring is still essentially a family task, which falls mainly on one or two people. Thus, the main caregiver would be the person, non-professional, family member, spouse or friend who provides most of the daily support to a person suffering from dementia [23].

The generalized involvement of activities of daily living (ADL) that dementia processes entail involves many tasks, time and dedication on the part of the caregiver. Communication between the person with dementia and the caregiver is also increasingly impaired, so understanding and making oneself understood is an extremely difficult task. This ‘time of care’ that the caregiver invests in the family member with dementia reduces the time for themselves and has been directly related to the levels of stress of the caregiver [24]. When people show a great impairment or chronic health conditions that seriously threaten their lives, many relatives experience anticipatory grief that generates feelings of denial, anger, depression and, finally, acceptance of reality. They progressively experience the loss of their loved one with the consequent conflicting feelings. For some family members and caregivers, the feeling of grief and its negative repercussions are even more intense in anticipatory grief than in post-death grief [25,26].

Anticipatory grief [27] is the term used to refer to this particular moment that is described as the emotional response to the potential threat of the death of a loved one or of oneself. Anticipatory grief does not imply a completion of a part of the grieving process when the person dies, but anticipating a death allows the individual to understand the loss as a natural process and to unfold their coping mechanisms to make it less painful [28]. Anticipatory grief, therefore, is an active psychological process of thoughts and emotions, very different from the mere anticipation of death, which involves three fundamental questions [27]: the feelings inherent in grief and acceptance of losses (distress, anxiety, sadness); individual and family reorganization (roles, unfinished matter); and the slow detachment and facilitation of a proper death.

### Purpose of the Present Study

The purpose of the present study is to determine which characteristics of caregivers are related to the presence of anticipatory grief. At the moment, few studies have focused on determining the characteristics of caregivers of family members with Alzheimer’s disease and other dementias; however, they are the main caregivers, so they have particularly harsh living conditions and numerous healthcare needs. 

With this purpose, the present study focuses on determining the impact of certain characteristics such as age, gender, level of education, relationship with the person with dementia, years with dementia, type of diagnosis or number of years as a caregiver on the presence of anticipatory grief. On the other hand, the current study intends to assess other caregiver’s psychological outcomes, such as caregiver burden and psychological symptoms, trying to explore their association with anticipatory grief. 

In this sense, and regarding the variables of significant impact, we hypothesize that a longer duration of the disease (advanced and severe phases) increases the likelihood of anticipatory grief in the family caregiver [5,26]. Likewise, we expect the relationship with the person with dementia, spouses or children to influence the results of anticipatory grief [29]. In the same way, the existence of psychopathology in the caregiver is expected to hinder the acceptance of losses and increase the elements of burden, sadness and isolation typical of anticipatory grief [30]. Additionally, it is expected that caregiver burden significantly correlates with anticipatory grief [31]. Finally, and after all, we hypothesize that all these variables can predict anticipatory grief.

## 2. Methods

### 2.1. Participants

The participants that compose the sample of the current study are caregivers of relatives with dementia. An initial sample was intentionally recruited from direct contact with the families. With a confidence level of 95% and under the assumption of maximum indeterminacy, the maximum sampling error was 0.04331. 

We excluded from the study all those participants with some cognitive or language comprehension difficulty that was determined to decrease the validity of the answers. In the same way, those participants who could be particularly emotionally affected and answer to the battery of tests that could negatively affect their state of mind were excluded. Based on these exclusion criteria, only two people were excluded from the initial sample (1.5%) because they themselves decided not to participate when reading the questionnaire. one of them was due to a recent loss and the other because she did not seem able to continue. 

### 2.2. Measures

Sociodemographics and caregiving experience. For the purpose of this study, an ad hoc demographic information sheet was administered. It asked about age, gender, educational level and employment status, family structure, marital status and family relationship with the relative, diagnosis, characteristics of the care experience and possible consequences derived from it. All this information was collected based on the significant associations previously found in the references.

Anticipatory Grief. The Marwit Meuser Caregiver Grief Inventory (MM-CGI) [32] is a 50-item inventory designed to measure the grief experience of family members who are caregivers of people living with a diagnosis of neurodegenerative dementia (e.g., Alzheimer’s disease). The inventory provides information on three relevant factors in the measure of anticipatory grief (personal sacrifice burden, feelings of sadness and nostalgia, worry and isolation) as well as a total grief score. ‘Personal sacrifice burden’ includes individual losses experienced as a result of caregiving, such as an impairment in physical health or loss of personal freedom; ‘feelings of sadness and nostalgia’ measures intrapersonal emotional reactions in response to caregiving; and ‘worry and isolation’ represents feelings of losing connections with and support from others, as well as worries about future losses. The MM-CGI has shown good psychometric properties, both in reliability and validity, with high scores in internal consistency (α = 0.90 and 0.96). The translation of the MM-CGI into Spanish has been carried out on the basis of this study, with the consent of the authors. The questionnaire was translated by psychologists into Spanish, and the back-translation was carried out by a British native (percentage of agreement between the items: 100%). 

Caregiver Burden. The caregiver burden was assessed from two measures: the Zarit Burden Interview and the Spanish version of Caregiver Strain Index. The first is a self-report questionnaire that assesses the degree of subjective burden of caregivers of the elderly or other dependent persons. The questionnaire, adapted to Spanish by Martín et al. (1996), consists of 22 items; each item is scored on a Likert scale gradient of 5 points. The questionnaire has values of 0.91 for internal consistency and 0.86 for test–retest reliability [33].

With regard to the Spanish version of the Caregiver Strain Index (CSI) [34] validated by [35], it is a self-report measure composed of 13 items with true–false dichotomous response options that assess the degree of overexertion of caregivers. At the psychometric level, the Spanish version presents an adequate measure of internal consistency (0.81), and its criterion validity and concurrent validity have been studied with the Duke social support scale [36,37], a scale of dependence on Barthel’s Activities of Daily Living [38] and the anxiety and depression scale questionnaire [35,39].

Psychopathology. To assess the presence of psychological symptoms, three measuring instruments were administered. The Derogatis Symptom Checklist, Revised, SCL-90-R [40], is a self-report instrument to assess the degree of current psychological distress. It consists of a list of 90 psychopathological symptoms of varying levels of severity (range of 0–4 points. The reliability of the scale is very acceptable, with internal consistency ranging between 0.77 and 0.90, depending on the scale and study [40,41,42,43]. The Spanish version of the inventory used in this study was developed by Casullo [44].

On the other hand, the Depression, Anxiety and Stress Scale is the short form (21 items; DASS-21) of the original scale by Lovibond and Lovibond [45] that is composed of 42 items for self-reported assessment of depression, anxiety and stress. The Spanish version was translated by Daza [46]. The DASS-21 is composed of three Likert-type subscales with four response points. The reliability, evaluated through Cronbach’s α, has also been shown to be acceptable for the three scales of depression, anxiety and stress (0.81, 0.73 and 0.81, respectively [45].

Finally, with the aim of being able to assess the possible link of depressive symptoms in caregivers with the experience of anticipatory grief, the Patient Health Questionnaire (PHQ-9) was administered. The PHQ-9 is a diagnostic tool developed by Spitzer et al. [47] composed of nine items that evaluate the presence of depressive symptoms (corresponding to the DSM-IV criteria) present in the last 2 weeks. At the psychometric level, the PHQ-9 presents good reliability (α = 0.89) [47] and adequate criterion validity (sensitivity of 0.80 and specificity of 0.92) [48].

### 2.3. Procedure

The present study used a cross-sectional design and followed a non-probabilistic sampling from different associations in Catalonia (Spain). First, contact was made with the coordinators of different associations of relatives and day care units for people with dementia and their families to inform them of the purpose of the study and ask them to participate as a center to develop the research. All of the entities consulted decided to participate in the investigation, providing their verbal consent and their collaboration when transferring the information to the possible participants. Subsequently, sessions were held to inform those caregivers about the purpose of the research and to require the subjects’ voluntary participation. They were informed of the confidential nature of the data and of their ability to leave the study if it caused them a high level of discomfort.

After the verbal consent of the entities and the written consent of the relatives was obtained, in mid-2017, the questionnaires were distributed. The administration of the questionnaires was completed taking into account the order of the instruments and the adequate time to avoid fatigue. In addition, each participant was individually monitored by trained psychologists. Thus, the understanding of the questionnaire was ensured, and any doubt or difficulty could be resolved during its completion. 

### 2.4. Data Analysis

The statistical analysis of the data was carried out using the statistical package IBM SPSS Statistics version 27 (Statistical Package for the Social Sciences, Chicago, IL, USA) for those analyses in which usual inferential techniques based on parametric or nonparametric reference distributions were used. In the basics contrasts, techniques based on Student’s *t*-test (comparison dichotomic categorical variables as gender or type of dementia), Snedecor’s F (comparisons of multigroup as marital status or educational level), estimates of Pearson correlations for linear bivariate distributions and Spearman’s correlations for nonlinear ones have been used (estimation of the relationship between quantitative variables). In addition, and to describe the impact of the different quantitative measures on the observed distribution of the anticipatory grief, we used the linear stepwise regression with an inclusion criterion of *p* < 0.001. In general, we determine a signification criterion of *p* < 0.001 to reduce the probability of type I error, according to the Bonferroni recommendations.

## 3. Results

### 3.1. Demographic Variables Associated with the Care Process

The sample consisted of a total of 129 subjects, mostly women (67.8%), aged between 32 and 85 years (M = 62.09, SD = 10.89), who were married (82.5%) and had primary or secondary education (66.1%), and who had been caring for a family member with dementia, in most cases with an Alzheimer’s disease diagnosis (65.1%), that was between 0 and 10 years of evolution (86.3%). In general, the assessed caregiver was the main caregiver of the relative with dementia (70.7%); the caregivers were normally children (61.3%) and spouses (32.8%). In most cases, the family member requiring care lived in the caregiver’s home (51.7%). The majority of caregivers spent 7 days a week caring for their family member (55.0%), and it was usual for them to be the only caregiver of the family member (53.1%) and without the assistance of a professional caregiver (70.0%). In addition, some of the caregivers usually have other care responsibilities (33.1%). In this sense, most caregivers combine care hours with hours of rest (79.0%). However, a considerable percentage show a loss of independence (72.6%), fatigue (73.0%) and feelings of guilt if they are not taking care of the family member (44.1%). Table 1 shows the general description of the categorical variables.

The average age of the sample of caregivers was 61.56 years, and that of the persons with dementia was 82.21 years. The assessment occurred on an average of 6.59 years since the diagnosis was made. Regarding care time, caregivers reported an average of 65.74 h a week dedicated to caring for the person with dementia and an average of 6.80 years performing this care function (see Table 2). As it is an ex post facto study, the caregivers’ assessments were made from already consolidated diagnoses of people with dementia. Therefore, the clinical diagnosis already described in the patients’ medical history was directly assumed. To avoid confounding effects between diagnoses, they were grouped into three broad categories of Alzheimer’s, MCI and others, clearly independently, to eliminate the possibility of confounding clinical classification.

On the other hand, in the case of the people with dementia, most of them were married (48.3%) or were widowed (46.7%), had between two and three children (61.6%), had either a primary education level (38.3%) or were without studies (45.8%), were usually retired (49.2%) or were housewives (42.5%) and had a medium socioeconomic level (45.8%). Approximately half of them lived with their caregiver (48.3%).

### 3.2. General Psychological State of Caregivers

The reliability analysis of each of the administered instruments included Cronbach’s α values between 0.734 (Caregiver Strain Index, CSI) and 0.969 (Derogatis Symptom Checklist, Revised, SCL-90-R).

The caregivers’ scores on the questionnaire about anticipatory grief were average values, both in regard to their individual dimensions of burden ($\bar{x}$ = 56.84; sd = 16.23), sadness ($\bar{x}$ = 48.23; sd = 13, 98) and isolation ($\bar{x}$ = 43.50; sd = 11.28), as well as in regard to the total score ($\bar{x}$ = 149.70, sd = 34.95). Regarding the level of strain and burden, the average of the caregiver’s scores did not indicate a high level of strain ($\bar{x}$ = 6.13; sd = 2.99) with respect to the Caregiver Strain Index, with the average being below a score of 7, nor with respect to the Zarit Burden Interview with scores lower than 46 ($\bar{x}$ = 28.69; sd = 16.66).

At the psychopathological level, the average scores were within the range described in the Spanish population for each of the SCL-90-R and the DASS-21 scales. However, the mean reported for the Patient’s Health Questionnaire may be indicative of possible depressive symptomatology ($\bar{x}$ = 16.74; sd = 6.79) (see Table 3).

### 3.3. Caregiver’s Characteristics Related to Anticipatory Grief

In the analysis of the correlations of the factors with the observed distributions of anticipatory grief, inverse relationships were established between the age of the people with dementia and the Burden (r = −0.301; *p* = 0.004; gl = 89), Sadness (r = −0.205; *p* = 0.050; gl = 92) and Isolation dimensions (r = −0.297; *p* = 0.003; gl = 96) and the total score of the MM-CGI (r = −0.296; *p* = 0.009; gl = 76). Table 4 shows these results. Nevertheless, there was no relationship between the age of the caregiver and any of the dimensions of anticipatory grief. Neither level of study determined significant differences. Conversely, regarding gender, the results indicated that females showed a higher average in the Sadness dimension of the anticipatory grief inventory than males (51.40 vs. 42.37) (*t* = 3.191; gl = 96; *p* = 0.001; r = 0.271).

With respect to the disease, the results about the diagnoses of the person with dementia showed a slight effect associated with lower scores in all dimensions when the person with dementia had mild cognitive impairment (Burden: F = 2.888 *p* = 0.061; Sadness: F = 1.436 *p* = 0.243; Isolation: F = 0.738 *p* = 0.481; Total: F = 2.936 *p* = 0.060). Appendix A of Appendix A shows this slight effect (ε^2^ between 0.156 and 0.312) with definition of the confidence interval of 95%.

On the other hand, there was no significant relationship between the duration of dementia and any of the dimensions of anticipatory grief (see Table 4). However, positive relationships were established between caring time (years) and Burden (r = 0.151 *p* = 0.154 gl = 91, Sadness (r = 0.320 *p* = 0.002 gl = 95) and Isolation dimensions (r = 0.247 *p* = 0.014 gl = 96) and the total score of the MM Caregiver Inventory (r = 0.228 *p* = 0.047 gl = 77). Likewise, positive relationships were established between the time dedicated to caring (hours) and the Burden (r = 0.264 *p* = 0.014 gl = 87), Sadness (r = 0.206 *p* = 0.050 gl = 91), Isolation dimensions (r = 0.223 *p* = 0.029 gl = 96) and the total score of the MM Caregiver Inventory (r = 0.253 *p* = 0.028 df = 75; see Table 4).

Regarding the relationship between person with dementia and caregiver, the results showed a positive relationship between being a spouse and the Burden (t = 2.610, gl = 87, *p* = 0.0055), Sadness (t = 2.627, gl = 90, *p* = 0.005) and Isolation dimensions (t = 3.683, gl = 93; *p* < 0.001) and total score (t = 3.426; gl = 73; *p* = 0.0005) of the anticipatory grief inventory, indicating that spouses usually have higher scores in these dimensions than children (ε^2^ between 0.212 and 0.416). In addition, the results showed a positive relationship between being the main caregiver and higher scores on the Burden (*t* = 3.261, df = 89, *p* = 0.001) and Isolation dimensions (*t* = 2.524, df = 96, *p* = 0.0053) of the MM-CGI (ε^2^ between 0.187 and 0.354).

Finally, Table 5 shown the results of the stepwise regression with restrictions of tolerance to avoid collinearity. We include the linear regression study established to detect the predictor variables of the four anticipatory grief scores. For this, the quantitative variables in Table 3 were included as regressors so that the partial regression coefficients represent the impact of each of these variables on the prediction of the anticipatory grief components. Given the number of regressor variables, a stepwise inclusion criterion was chosen but with *p*-value levels lower than 0.001 to incorporate variables. This rigorous criterion was selected to avoid collinearity effects and maximize the explained variance of the observed distribution of anticipatory grief components. The results are clearly demonstrative of the crucial paper of the distribution of the Zarit scores to predict the observed distribution of Anticipatory Grief in most of the factors. The Burden, Isolation and the total score are highly connected with Zarit scores.

## 4. Discussion

The purpose of the present study is to determine which characteristics of caregivers are related to the presence of anticipatory grief, hypothesizing that the variables that will have a particularly significant impact will be the relationship with the person with dementia as well as the duration of the disease because both can increase the probability of anticipatory grief in the family caregiver. Additionally, the study attempted to assess caregivers’ other psychological outcomes, such as caregiver burden and psychological symptoms. With all this, it is expected to be able to predict the existence of anticipatory grief in caregivers from the above variables.

On the first objective, related to the characteristics of the caregiver, the results obtained indicate that women are the most frequent caregiver profile of relatives with Alzheimer’s disease and other dementias. Acquiring this task of caring is not something that happens randomly. Although there is a generalized thought that the care of members with chronic or dependent diseases is the responsibility of the family, the responsibility usually falls on women. There are different variables that can determine this decision; for example, the caregiver gender, the presence of dependent children in the family, the position of the caregiver within the family, the caregiver’s age, the caregiver’s availability of time, etc. [24,49]. In addition, in some countries, the sense of duty toward the family is so internalized that it remains the main system for the provision of care and well-being, and for some population groups, family is the only possible system for care. On the other hand, there is a social construct that women fill the role of caregivers based on beliefs about the protective functions of the family and of women as natural care providers [50].

With respect to the caregiver relationship, spouses have a higher level of anticipatory grief than other family members, such as children. This finding agrees with the results of previous studies with higher scores in all the dimensions of anticipatory grief [31]. Wright [29] pointed out that the special situation that spouses live with compared to other close relatives, such as daughters or other family members, should be differentiated. The irruption of the disease is affected by specific aspects of the marital relationship. In fact, the stress inventory of Holmes and Rahe [51] had attributed a very high score to the loss of a spouse. We also have reported an interesting observation regarding the evolution that different emotions follow in different groups of caregivers, children and spouses [5]. Grief, anger, concern about certain issues of the past, present or future, etc., change at each stage of the disease. Risk indicators in psychopathology favor greater probabilities of anticipatory grief, especially in the Burden and Sadness dimensions. Antecedents related to some of the forms of depression and the existence of unresolved grief in the past are two of the main features of the risk profile. This could account for the results obtained when assessing the presence of depression in some caregivers as part of the second research objective of the present study.

In relation to the age variable, in the results obtained, the age of the caregiver does not seem to influence the appearance of the symptoms of anticipatory grief. However, the older the person with dementia is, the greater the tolerance of deterioration and the less the burden on caregivers is. Likewise, the duration of the disease in absolute terms does not seem to influence the onset of symptoms of anticipatory grief. Despite this, Holley (2009) warns that the dementia process is ‘special’ because it is an ‘accumulation of losses’. This accumulation, in relation to the grieving processes, makes it complicated. In this sense, the results obtained indicated that the time dedicated to care is more important, in a chronological sense and, above all, in terms of day-to-day intensity, correlating positively with symptoms of burden, sadness and isolation. In fact, it is logical to think that as the time dedicated to care increases, the leisure time and social activity decreases, and at the same time, the probability of physical and psychological symptoms increases, all of which affect the perception of burden [52]. These two concepts, time and symptoms, do not maintain a causal relationship but a contingency. Alzheimer’s disease usually has an uncertain duration, from 1 or 2 years to 20 or more [53]. Proposing a long-term effort without limits and without a clear organization of resources entails a risk of burden and claudication that these results confirmed. Therefore, having information and early specialized support is of vital importance to family caregivers. 

Likewise, a second, more relevant aspect related to time is the dedication (as intensity) to the care tasks from a day-to-day agenda perspective. The results indicated that a greater dedication to care, without support, limits or relational compensation (leisure, friendships, etc.), causes a greater problem of burden, sadness and isolation [52]. Both concepts, time and symptoms, can also be related to the phase or type of dementia and therefore to the caregiver dependence and burden, giving rise to different reactions of anticipatory grief, as some studies have determined [5,21,26], which in turn can also be related to the results obtained when differentiating the type of dementia diagnosis, with lower scores in all dimensions of anticipatory grief when the person with dementia had mild cognitive impairment with respect to other diagnoses.

Finally, from the regression analysis carried out from the variables that assess both the caregiver’s burden and psychological symptoms, it can be inferred that the Zarit measure of caregiver burden is the variable that best predicts anticipatory grief. The results are clearly demonstrative of the crucial role of Zarit scores to predict the observed distribution of anticipatory grief. In this sense, it is known that the operating variables in the burden symptoms are multiplied by the effect of the different variables studied, giving rise to the appearance of what we call overload. The translation to elements classified as anticipatory grief will allow an attentive approach to the caregiver’s needs throughout the entire disease process: improving the quality of care, reducing the repercussions on the caregiver’s health and facilitating the post mortem grieving process [19]. Thus, the different programs aimed at family caregivers, both those aimed at informing in the initial stages and those focused on training and/or later support, favor the management of emotions related to loss and grief as a priority objective [32]. Although these results shed light on all the variables involved in the care task, they highlight the need for multidimensional theoretical models that capture the complexity of all the variables involved [54].

## 5. Limitations

First, the cross-sectional approach used to obtain information prevents us from establishing causal relationships between variables. However, the present study can be used as a guide to better understand the role of caregiver characteristics in the relationship with anticipatory grief. Second, it is worth mentioning the size of the sample, which requires a certain prudence in the generalization of the results. Additionally, all the participants were linked to family associations, which leaves out people who do not have any support; following the result trends, this aspect could increase the percentage of severity in terms of anticipatory grief in other samples with fewer opportunities for support and attention. Another limitation of the sample, although it reflects the social reality, is the high number of women participants.

However, it should be taken into account that the variability in the diagnosis date was broad, although most people with dementia had been diagnosed between 6 and 10 years prior. In relation to this variability, it is worth noting that the emotional ups and downs of the caregivers throughout the disease process, which can be treated therapeutically, make it difficult to measure these characteristics with self-report and cross-sectional questionnaires. In the same way, a significant percentage of the caregivers reported fatigue and discomfort in the face of changes; however, their relationships with the person with dementia changed constantly. Thus, for future research, it is suggested that the caregiver characteristics are measured at different times of the disease.

Finally, the results of the current research have not included any mediation study, and our results do not allow it to be presented. Further research is required to provide evidence of this point.

### Practice and Policy Implications

It is very important to establish therapeutic strategies oriented toward working with caregivers [19]. First, in regard to their perception of life, not so much from a philosophical perspective but from the vital organization (free time, friendship relations, social participation, etc.) aspect. Second, in regard to their relationship with the person with dementia from a new perspective. Third and finally, to increase caregivers’ competence assessment and their ability to cope with stressful, frustrating and even threatening situations. The evidence shows that each subject is different, so that each person with dementia and each family caregiver constitute a different complex reality. However, there are a series of factors dependent on the caregiver that can contribute to increasing the feeling of discomfort or overload or, on the contrary, act as protective factors of said feelings [52,54]. Knowing the most frequent caregiver profile of relatives with Alzheimer’s disease and other dementias allows defining plans and intervention strategies and community attention focused on their needs.

As Sanders (1980) points out, the concept of motivation for change is very important and, fortunately, it is trainable. This is a good idea for planning possible interventions of stress management with family caregivers [55,56]. Another aspect that acquires special relevance is the resolution of ‘pending’ issues. It is recommended that caregivers address these issues as soon as possible since deterioration often makes them impossible. The management of emotions by the caregiver is very important, especially for its relationship with possible feelings of guilt and the repercussions that this can have both emotionally and in the care task. 

## 6. Conclusions

Anticipatory grief is considered an active psychological process of thoughts and emotions that is very different from the mere anticipation of death. The results obtained show that time dedicated to care correlates positively with symptoms of burden, sadness and isolation, fundamental constructs of anticipatory grief. 

Regarding the profile of the caregivers, the results indicate that the spouses have a higher level of anticipatory grief than children. Furthermore, a long-term effort without limits and without a clear organization of resources entails a risk of burden and claudication, which proves to be one of the greatest predictors of anticipatory grief. Thus, having information and early specialized support that focuses on managing emotions related to loss and grief is of vital importance to family caregivers.

## Figures and Tables

**Table 1 ijerph-18-08838-t001:** Sociodemographic characteristics—categorial variables.

		*n*	%
Gender	Male	38	32.2
	Female	80	67.8
Marital status	Single	8	6.7
	Married	99	82.5
	Separated	1	0.8
	Divorced	9	7.5
	Widower	3	2.5
Educational level	None	5	4.2
	Primary	33	28.0
	Secondary	45	38.1
	University students	35	29.7
Employment situation	Active worker	43	36.1
	House chores	26	21.8
	Retired	49	41.2
	Unemployed	1	0.8
Where the person with dementia lives	With caregiver	61	51.7
	Without caregiver	9	7.6
	At home assisted	8	6.8
	In residence	38	32.2
	Others	2	1.7
Type of dementia	Alzheimer	84	65.1
	Other dementias	23	17.8
Years since diagnosis	0–5	57	55.9
	6–10	31	30.4
	11–15	6	5.9
	16–20	6	5.9
	More than 20	2	2.0
Main caregiver	Yes	82	70.7
	No	34	29.3
Caregiver’s relationship	Spouse	39	32.8
	Child	73	61.3
	Grandchild	1	0.8
	Sibling	2	1.7
	Other: niece, nephew	4	3.4
Weekdays dedicated to caring	0	2	1.6
	1	2	1.6
	1.5	1	0.8
	2	5	3.9
	3	11	8.5
	4	11	8.5
	5	6	4.7
	5.5	1	0.8
	6	6	4.7
	7	71	55.0
	Alternating days	1	0.8
	Alternating weeks	1	0.8
Other care responsibilities	Yes	39	33.1
	No	79	66.9
How many people care for the person with dementia	0	16	16.7
	1	51	53.1
	2	15	15.6
	3	8	8.3
	4	5	5.2
	6	1	1.0
One caregiver is a professional caregiver	Yes	33	30.0
	No	77	70.0
You combine care hours with hours of rest	Yes	83	79.0
	No	22	21.0
Guilt if you are not caring	Yes	49	44.1
	No	61	55.0
Loss of independence	Yes	85	72.6
	No	32	27.4
Fatigue	Yes	84	73.0
	No	31	27.0
Do you often feel like quitting?	Yes	18	16.7
	No	90	83.3
Do you think that a residence would be better if it was possible?	Yes	46	45.1
	No	56	54.9
Do you often think that there is no one better than you to care for the person with dementia?	Yes	63	54.3
	No	53	45.7
Do you think that, if you are not present, unexpected negative things can happen?	Yes	49	42.6
	No	66	57.4
Household income per month	Up to 1000 EUR	77	59.9
	From 1001 to 2000 EUR	37	28.7
	More than 2000 EUR	15	11.4

**Table 2 ijerph-18-08838-t002:** Sociodemographic characteristics.

	Min.	Max.	$\bar{x}$	sd
Caregiver age (years)	32	85	61.56	10.78
Person with dementia age (years)	61	100	82.21	8.58
Number of years since diagnosis	1	37	6.59	8.58
Time spent caring for the person with dementia (h/month)	4	200	65.74	67.60
Time spent taking care of the person with dementia (years)	1	21	6.80	4.83

Note. $\bar{x}$ is the arithmetic mean of the observed distribution and sd is the standard deviation.

**Table 3 ijerph-18-08838-t003:** Scores on the psychometric tests by the sample of caregivers.

	Min.	Max.	$\bar{x}$	sd
Anticipatory Grief—Burden Dimension	18	85	56.84	16.23
Anticipatory Grief—Sadness Dimension	15	75	48.23	13.98
Anticipatory Grief—Isolation Dimension	18	72	43.50	11.28
Anticipatory Grief—Total Score	58	231	149.70	34.95
Caregiver Strain Index—CSI Total Score	0	13	6.13	2.99
SCL90-R Somatization	0	36	10.90	8.45
SCL90-R Obsessions	0	30	11.63	7.38
SCL90-R Interpersonal Sensitivity	0	30	4.94	5.31
SCL90-R Depression	0	45	14.38	9.73
SCL90-R Anxiety	0	21	7.67	5.48
SCL90-R Hostility	0	12	3.12	2.69
SCL90-R Phobias	0	13	2.65	3.04
SCL90-R Paranoid	0	19	2.93	3.60
SCL90-R Psychoticism	0	24	3.79	4.20
SCL90-R Additional	0	22	6.29	4.87
SCL90-R Total Score	0	223	68.10	47.72
PHQ-9 Total Score	0	32	16.74	6.79
DASS-21 Anxiety	0	30	5.06	6.95
DASS-21 Stress	0	38	10.49	9.63
Zarit Burden Interview Total Score	1	69	28.69	16.66

Note. $\bar{x}$ is the arithmetic mean of the observed distribution and sd is the standard deviation.

**Table 4 ijerph-18-08838-t004:** Correlation between the dimensions of the anticipatory grief inventory and sociodemographic variables (patient’s age, duration of illness and caring time).

	Burden	Sadness	Isolation	Total
Burden	1			
Sadness	0.648 **	1		
Isolation	0.667 **	0.596 **	1	
Total	0.903 **	0.856 **	0.828 **	1
Age of caregiver	0.050	−0.011	0.107	0.100
Age of people with dementia	−0.301 **	−0.205	−0.297 **	−0.296 **
Diagnosis years diagnóstico	−0.109	0.076	0.020	−0.057
Care hours	0.264 *	0.206	0.223 *	0.253 *
Care years	0.151	0.320 *	0.247 *	0.228 *

* *p* < 0.05 ** *p* < 0.01.

**Table 5 ijerph-18-08838-t005:** Parameter estimation of the stepwise linear regression to predict Anticipatory Grief.

Criteria Variable	Best Model	Parameter	Standard Error	*p* Value	Holmes *R*^2^
AG Burden Dimension	Zarit Total Score	*B*_1_ = 0.529	0.088	<0.001	0.405
AG Sadness Dimension	CSI Total Score	*B*_1_ = 1.954	0.507	<0.001	0.250
AG Isolation Dimension	Zarit Total Score	*B*_1_ = 0.449	0.069	<0.001	0.437
AG Total Score	Zarit Total Score	*B*_1_ = 1.016	0.211	<0.001	0.341

Note. AG has been used as an abbreviation for Anticipatory Grief.

## Data Availability

The data presented in this study are available on request from the corresponding author.

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
