# Peer review of "Alzheimer’s Disease Caregiver Characteristics and Their Relationship with Anticipatory Grief"

_ijerph, 2021, doi:10.3390/ijerph18168838_

Round 1
Reviewer 1 Report
Thanks for inviting me to review this manuscript, which topic is very important yet less investigated. In general, a better differentiation among the concepts is expected: (e.g. anticipatory grief vs complicated grief, AG vs burden/depression). There are a number of concepts investigated without being tightened by a conceptual framework, that weakened the values of this study.
I have some further suggestions for the authors to consider for an improvement.
Abstract
- The first sentence is not presenting accurately. Ambiguity and ambivalence are two aspects relate to the feelings of caregivers. Not the aspects of Alzheimer's disease.
- Please make sure if you are focusing on the population of caregivers of Alzheimer's disease or dementia. Also, be consistent with the Methods section.
- The results are not clearly presented.
Introduction
- As anticipatory grief is the primary focus of the topic. The authors may consider explaining what is anticipatory grief in the first paragraph.
- The authors elaborated that anticipatory grief of caregivers of dementia is characterized by ambivalence and ambiguous feelings. However, how the dependent variables to be studied are related to ambivalence and ambiguity is relatively less explained.
- Page 2, 2nd paragraph: from the title sentence of this paragraph, I would expect the risk factors and protective factors listed (e.g. affective dependence, competency in the management) are those of "complicated grief". If not, please differentiate the associated factors of complicated grief and anticipatory grief, which is the primary focus of the study. As AG is the primary focus, I am puzzled why complicated grief is mentioned without a clear explanation.
- What is the potential relationship between the independent variable (anticipatory grief) and the dependent variables (socio demo, burden, psychopathology)? This would be the ground of the hypotheses.
- The knowledge gap should be identified that could be addressed by this study.
- The purpose of the study should be in 2-3 sentences instead of a long paragraph, which is not really readable.
- I am not sure, but I suspect there is more than one objective. The primary objective is to identify the sociodemographic risk factors, and the second objective is to explore the association of the AG with the other psychological outcomes (e.g. burden, psychological symptoms etc).
Methods:
- Please follow a guideline in reporting, e.g. STROBE, that would facilitate the reading.
- Participants: Just report the eligibility criteria and no need to report the number of subjects recruited, their age etc., which should be written in the results section.
- How is the sample size estimated?
- Measures: it is unclear which one is the dependent/independent variable.
- Please explain why ZBI and CSI were used to measure burden simultaneously? They are highly similar. When you put them into the model for testing, how is multicollinearity? Similar situation for the SCL, PHQ and DASS.
- What is the date of the data collection period?
- Pg 5: A number of statistical tests were listed. Please state what was the purpose of these tests. In the other words, why these tests should be done? If the test(s) is/are for addressing the objectives (i.e. identifying the association of the number of variables with AG, regression is good enough, why do you need to conduct the T-test, F test, correlation etc).
- What is the sequence of the stepwise regression? Justify too.
Results
- Table 1: weekdays dedicated to caring; how many people care for the person with dementia are not in equal intervals. Do "alternating days" means 3.5 days/week? how to define the loss of dependence? how to define fatigue?
- Table 2: quantitative variables mean what? do you mean continuous variable? As the variables listed in Table 1 are quantitative too. what is the meaning of "s", does it mean standard deviation? Please be consistent with the use of acronyms over the whole manuscript. The maximum time spent taking care of the person with dementia was 21 years! Have you checked if the caregivers took care of the patient because of other diseases before dementia was diagnosed?
- Why did the reliability analysis of the instruments are the key findings and reported in P.7? Did the AG scores higher or lower than the threshold (please refer to the cut off scores of the original scale)
- Table 3: what is IEC, Social Support scale, SCL, Health Total, Rosemberg total? This table is totally not readable. The information cannot be found in the main text. As the primary objective of this study is to identify caregivers' characteristics associated with AG, Shouldn't be the results "Caregivers' characteristics related to AG" reported first?!
- I cannot read Figure 1.
- Sorry, table 5 is not understandable to me. Putting the sub-score and total score of a scale; or putting two different scales which are highly similar, into a model for testing is definitely problematic. I am afraid the analysis has to be rectified.
Discussion:
- Identifying caregivers' demo characteristic related to AG seems not very meaningful as these factors are not modifiable. Please highlight the implication of the study.
- Paragraph 1: highlight what knowledge gap you have addressed. Or recap what are the key findings.
- Limitation: selection bias? as only associations and daycare units are approached. The participants recruited there should have received much attention.
- What are the scientific values of this study? please highlight.
Reviewer 2 Report
The subject of the article is extremely interesting, and it can benefit the caregiving of people with dementia. The sample is not large, as you say at some point in the paper, but we all know how difficult it is to enroll caregivers in a study, as they have particularly hard living conditions. My congratulations for the work done.
However, it would be necessary to make some adjustments. Below, you can read my comments after carefully reviewing your paper:
- Page 1, line 14-15: The Abstract starts with a sentence about Alzheimer disease and two fundamental aspects: ambiguity and ambivalence. Per se, it is not possible to understand what it means.You have to take into account that the reader usually starts with the abstract, and it must be auto comprehensive. You should clarify the two fundamental aspects in relation with…
- The paper has in its title three issues: Alzheimer Disease, Caregiver Characteristics and Anticipatory Grief. However, the Introduction section is biased towards grief. I would recommend to include some background about the other two issues.
- About Caregiving, it would be appropriate to define what kind of caregiver are we talking about in the Introduction section. Are they professional caregivers? Formal or informal? Are they always family caregivers? The type of caregiving imposes some features to the burden, and it would be interesting to take it into account. Moreover, in the Introduction section we can find some interesting information about grieving, but nothing is said about caregiving. Only in the Purpose of the Present Study subsection, we find “characteristics of caregivers” as an issue.
- Page 2, line 53: Correct Sanders with capital letter.
- Page 2, lines 57-61: “... it is possible to differentiate between risk factors, … and lack of symptom control… and economic problems.” The sentence is so long, that it is not clear which is the second term when you mention “between”… I would suggest writing it down again.
- Page 2, lines 72-73: How to interpret this sentence? I don’t see the link between this sentence and the previous paragraph, and I consider that it should be necessary to give some more explanation, or delete the sentence.
- Page 2, line 76-77: The sentence is incomplete, or delete “If”.
- Throughout the article, authors write “Alzheimer disease”, “family members with cognitive impairment”, “person with dementia”, “chronically ill person”… The disease we are talking about should be clarified, because, even though they are similar, they are not the same.
- In the Measures section, you include as Sociodemographics the caregiving experience (“characteristics of the care experience and possible consequences derived from it.”), when it is not.
- Page 3, line 143 and line 145, correct verb form.
- The different evaluation instruments are well described, however, I see a problem. It seems that two different questionnaires have areas of overlap. That means we are measuring the same thing twice. Concretely, we measure depressive symptoms with the PHQ-9, and then with the DASS-21. Although one test is different from the other, the evaluation should be optimized so as not to fatigue people. If authors have a good justification to use the PHQ-9, the reason for overlapping evaluations should be very well justified.
- Page 8, line 283: Modify the format of the table’s title and the letter source.
- In the two first paragraphs of the Discussion section, I don’t see the continuity between the two.
- Table 1 has some problems with the English. For instance, Unemployment (instead of Unemployed), Without Caregiver (¿?), Son (instead of Child), or Brother (instead of Sibbling), etc.
- Figure 1 is missing.
- In the Discussion section, we find in the first paragraph some content related to the research that should be replaced in the Background section.
- The Discussion section should discuss the implications of the findings in context of existing research, but the link between them is not explicit. In fact, many findings we have seen in the Results section are not discussed.
- In the Discussion section, you should choose the results that are important and “discuss” their implications.
- It is relatively frequent to read statements of the Conclusion that reach far beyond the study design and results. But in this case, it is just the opposite. The study is very good, but the authors under-interpret findings. I would suggest expanding this section.
Author Response
Please see the attachment.

This manuscript is a resubmission of an earlier submission. The following is a list of the peer review reports and author responses from that submission.
Round 1
Reviewer 1 Report
Anticipatory grief is an important issue in caregiving in most chronic illnesses and also in dementia. A limited number of previous contributions attempted to define the determinants of AG in dementia. Unfortunately, most of them are not taken into account (Garand et al., ADAD 2012, Ki Cheung et al., BMC Palliat Care 2018, Holley and Mast Gerontologist 2009). These studies indicated a series of predictors including disease severity, subjective caregiver burden, high depression levels in caregivers, and spouse function.
The authors performed a cross-sectional study including 129 caregivers (67.8% women) of patients with dementia. Cases with negative emotions ( ? how did they detect them) have been excluded from the study introducing a major selection bias. No information are given about the clinical diagnosis and severity of cognitive decline, yet a reference to a slight decrease of AG scores is mentioned in mild cognitive impairment. The assessment of AG has been made using the MM-CGI questionnaire; caregiver burden was assessed using the Zarit Burden Interview and Caregiver Strain Index. The Derogatis Symptom Checklist, PHQ-9 and DASS-21 were administrated to evaluate the level of psychopathologiy among caregivers. Statistical analysis is poorly explained. Group comparisons and correlation analyses were performed but without correction for multiple comparisons (Benjamini Hochberg) despite the impressive number of comparisons made. In addition, multiple regression models are not included in the final analysis, a necessary step given the interdependence of most predictors. Strangely enough, the authors provide the descriptive data for the caregiver burden and psychopathology but did not use these variables in their correlation analyses. Overall, older age, being female, spouse function, higher caring time (both in years and per day) were related to AG among caregivers. In respect to age, the statement is contradictory (see lines 252 to 257).
Despite the valuable effort in a very important and rarely studied issue, this contribution suffers from methodological flaws. The exclusion of emotionally affected cases is counterintuitive and, in fact, excludes the most relevant proportion of cases in terms of AG (according to the previous contributions). Multiple comparison control is mandatory (and would radically change the significance of their data). Regression models inspired by the previously identified predictions should be added.
Reviewer 2 Report
Thank you for this interesting paper describing influential factors of anticipatory grief in caregivers. I felt that this paper has good potential and may be useful to researchers in the dementia care field, but requires some revisions.
- More results could be described in the abstract - such as statistical values.
- Throughout the introduction, more referencing is required.
- You talk about Alzheimer's disease in the introduction, but you don't explain what it is or why it may lead to anticipatory grief. This would be helpful to the reader. You also carry out this study in people "with cognitive impairment" rather than just Alzheimer's - maybe it would be good to explain why this is.
- You should use dementia friendly language throughout - especially around words such as "patient". Here is the link to the Australian guidelines. https://www.dementia.org.au/sites/default/files/resources/dementia-language-guidelines.pdf
- The introduction was hard to follow in places - language could be more concise to support the reader's understanding.
- Methods: How did you define a "person with cognitive impairment"?
- How did you assess carers' cognitive and language comprehension and emotion affectiveness?
- Is psychopathology the correct term - this is a little confusing - psychological status may be a clearer terminology.
- The data analysis section should be much more detailed.
- Results: How was Alzheimer's diagnosis confirmed?
- Should you have used correlations to look at the influence of being a spouse/child on grief, or would t-tests/non-parametric equaivalent be more appropriate here?
- Discussion: More references required throughout.
- Results should be explicitly linked to as the discussion is hard to follow in this regard
- What do you think is the impact of different dementia subtypes on grief?
- Is there any mediating factors for grief such as co-morbidities/health inequalities?
Reviewer 3 Report
This is an interesting study examining the characteristics of Alzheimer's disease caregiver and their relation with anticipatory grief. This is an interesting study. I have 5 concerns that are important for the authors to address 1. First, the writing for the current paper can be sharpen further. There are several sentences and paragraphs that are awkward and unclear. 2. The current study used cross-sectional design and this is a limitation that should be acknowledged in the limitation. 3. It will be great if the authors summarize the zero-order correlation in a table. That will allow the readers to understand the results easier. 4. It will be important for the authors to elaborate more regarding the theoretical implications of the study 5. More information regarding the exclusion criteria will be beneficial. For example, how did the authors determine who to exclude for those that could be particularly emotionally affected to the tests. How did the authors determine the language comprehension ability of their participants?Author Response
Please see the attachment.

Round 2
Reviewer 3 Report
I appreciate the authors' effort in revising the manuscript. There is some improvement. However, I still have some concerns regarding the data analysis:
- It is recommended to include standard error in Figure 1
- More clarification should be included for the stepwise regression presented in Table 5. The analysis was introduced without a proper explanation. Furthermore, some of the wordings are very awkward (e.g., signification).
Round 3
Reviewer 3 Report
The authors have sufficiently addressed my comments. However, it will be important for the authors to address the other reviewers' concerns before the acceptance of the current paper.